# Stress Management in Plants: Examining Provisional and Unique Dose-Dependent Responses

**DOI:** 10.3390/ijms24065105

**Published:** 2023-03-07

**Authors:** Mariyana Georgieva, Valya Vassileva

**Affiliations:** Department of Molecular Biology and Genetics, Laboratory of Regulation of Gene Expression, Institute of Plant Physiology and Genetics, Bulgarian Academy of Sciences, 1113 Sofia, Bulgaria

**Keywords:** higher plants, stress agents, dose–response relationship, bystander effect, genome instability, hyper-radiosensitivity, hormesis, transgenerational memory

## Abstract

The purpose of this review is to critically evaluate the effects of different stress factors on higher plants, with particular attention given to the typical and unique dose-dependent responses that are essential for plant growth and development. Specifically, this review highlights the impact of stress on genome instability, including DNA damage and the molecular, physiological, and biochemical mechanisms that generate these effects. We provide an overview of the current understanding of predictable and unique dose-dependent trends in plant survival when exposed to low or high doses of stress. Understanding both the negative and positive impacts of stress responses, including genome instability, can provide insights into how plants react to different levels of stress, yielding more accurate predictions of their behavior in the natural environment. Applying the acquired knowledge can lead to improved crop productivity and potential development of more resilient plant varieties, ensuring a sustainable food source for the rapidly growing global population.

## 1. Introduction

Climate change is a multifaceted phenomenon that affects plant and animal species, as well as their habitats and ecosystems due to altered weather patterns and an increased frequency of extreme weather events. It also contributes to the spread of pests and diseases [1]. Being rooted in one place, plants are highly sensitive to fluctuations in temperature, rainfall, radiation, and other environmental factors, which induce a range of short-term or medium-term reactions, such as the acclimation processes, or long-term phenomena, such as transgenerational adaptation [2]. These reactions can impact the physiological state of plants, affecting their growth and development and leading to reduced seed production and germination, decreased nutrient uptake, and water use efficiency, increased vulnerability to pests and diseases and, in extreme cases, death [3]. The kinetics of the normal biological response is contingent on the intensity and duration of the stressor (acute or chronic) [4]. Plants have developed intricate defense mechanisms to withstand natural adversities and secure their survival against various challenges [5]. Altering the anatomy and morphology of plants through modification of their structural features is a potential strategy for combating the consequences of climate change [6,7,8]. At the genetic level, plants have developed mechanisms to repair or tolerate DNA damage, allowing them to maintain genome integrity [3,9]. A recent review by Hartmann et al. [2] traces the impact of different stressors on the stability of plant genome, focusing on the crucial underlying mechanisms. They indicate that environmental stress can cause changes in plant DNA, including mutations, epigenetic modifications, and alterations in gene expression. These changes can have a profound impact on plant growth, development, and ability to adapt to changing conditions. The crucial role of the potential interplay between genetic and epigenetic elements in understanding plant stress responses has been discussed. The effects of environmental stimuli on plants can sometimes be atypical and multifaceted, requiring more comprehensive analysis.

This review endeavors to give an insight into current knowledge of plant genome stability, with particular emphasis on the provisional and unique dose-dependent stress responses that are essential in determining plant growth and development when exposed to low or high stress levels.

## 2. Principal Stressors Impacting Plants

Any extreme event or climate variation can disrupt the optimal conditions for plants, leading to reduced plant growth, leaf and root development, and reduced crop yield [4,5]. Plants are affected by a variety of stress factors, which can be grouped into two main categories: biotic and abiotic.

Biotic stress factors include pathogens, pests, and inter-species rivalry, which cause harm to the plant through infections, herbivory, and resource competition [10,11]. The growth and health of plants can be greatly impacted by various types of pathogens, such as bacteria, viruses, and fungi, leading to a wide range of symptoms, including wilting, discoloration, and even plant death. Pests such as insects, mites, and nematodes can also inflict significant damage to plants by feeding on plant tissue, causing harm to leaves, stems, and even roots, as well as transmitting diseases, thereby exacerbating overall harm to the plant [12].

Biotic stress can cause detrimental effects on plant growth and development, including but not limited to decreasing plant biomass, disrupting photosynthesis, and altering plant morphology [13,14]. This generally leads to the production of defense compounds, such as phenolics and alkaloids, which can negatively affect the nutritional quality of crops. The effects of biotic stress on plants depend on the type of stress, the severity of the stress, and the ability of plants to elicit appropriate defense mechanisms [15].

During biotic stress, plants activate various hormonal pathways to cope with the stressor [13,16]. The Jasmonic acid (JA) pathway plays a crucial role in plant defense against insect herbivores, necrotrophic fungi, and some bacterial pathogens. Upon perceiving the presence of these stressors, plants release JA, which triggers a cascade of signaling events, ultimately leading to the expression of defense-related genes and the production of specialized metabolites, such as phytoalexins, and protease inhibitors [15,17]. Salicylic acid (SA) is another major phytohormone that plays a vital role in plant defense against biotic stress. It triggers the expression of genes responsible for the production of pathogenesis-related proteins that directly attack the pathogen or restrict its spread [18]. Additionally, the host defense system can be reinforced by administering exogenous SA treatment [19]. The ethylene pathway is also activated in response to a range of biotic stressors, such as insect herbivory, pathogen infection, and mechanical damage. This results in increased ethylene levels and the production of defense-related proteins, secondary metabolites, and the activation of signaling pathways involved in plant defense [20]. The mentioned regulatory pathways involved in plant defense exhibit notable differences, yet they also intersect to offer protection against pathogens. Furthermore, phytohormones such as abscisic acid (ABA), auxin, brassinosteroids, cytokinins, gibberellins, and peptide hormones are also involved in modulating plant immune responses [15,21,22,23]. Among these, JA is particularly important in triggering the plant defense system and cross-talks with other phytohormonal pathways [24].

Abiotic stressors encompass a wide range of environmental factors, including water stress, temperature fluctuations, imbalanced nutrient and mineral levels, and exposure to various forms of radiation, such as non-ionizing and ionizing radiation (IR) (high and low linear energy transfer, LET), as well as ultraviolet radiation (UV). Water stress (drought and flooding) affects many aspects of plant growth and development induced by genetic and molecular changes, which alter various biochemical and physiological processes [25] and have both individual and community-wide effects on plants [26]. Temperature stress can result from exposure to extreme cold, freezing temperatures, or high heat, which damage plant tissues and disrupt growth [27,28]. All macronutrient deficiencies inhibit plant growth and development [29]. Exposure to non-ionizing and IR and ultraviolet radiation can harm plants through photoinhibition and photo-oxidation processes [30]. All abiotic stress factors usually lead to genotoxic and oxidative stress, which damages plant DNA and other cellular components [28,31]. Some abiotic stressors affecting plants have anthropogenic origin, namely ozone, carbon dioxide (CO_2_), emissions of sulfur and nitrogen oxides (SOX and NOX), formaldehydes, ammonia and various complex mixtures, soil pollutants such as soil salinity [4,32,33,34], and the use of nanoparticles in agriculture, such as nano-fertilizer and nano-pesticides [35]. In addition, plants have specialized mechanisms to counteract the effects of sulfur dioxide (SO_2_) from acid rain [36]. At low concentrations, SO_2_ can act as a signaling molecule in plants and play a role in seed germination, stomatal opening and closing, and plant response to various environmental stimuli [37]. Similarly, low concentrations of nitric oxide (NO) have a signaling role in plants. NO is non-toxic and has an impact on plant growth and development, senescence, ethylene production, resistance to diseases, and stress tolerance [38]. The combination of natural and anthropogenic factors leads to the spread of heavy metals (Cd, Cr, Pb, Al, Hg, etc.) in soil through volcanic eruptions, industrial activities, and modern agricultural practices. These elements play no role in fundamental metabolic processes and do not have major physiological functions in plants. All of these factors can trigger changes spontaneously, intentionally or as a result of metabolic activity. The diversity of biotic and abiotic factors in plants is illustrated in Figure 1.

## 3. Impact of Abiotic Stress Factors on the Plant Genome: Direct Effects

Environmental factors or treatment with agents can affect plant biological macromolecules (DNA, proteins, and lipids) in a direct or indirect manner, leading to oxidative, genotoxic, and cytotoxic effects on plant cells. These effects can cause genomic instability, mutation, carcinogenesis, or even cell death, which can negatively impact plant health and result in reduced crop yield [33,39]. The inability of cells to preserve the integrity of their own genome is linked to primary damage in their biomolecules. The nature of primary DNA damage is similar, regardless of it occurs in microbial, animal, or plant genomes. Whether it originates from external or internal sources, damage can be hydrolytic (cleavage of glycosidic bonds, deamination of cytosine analogs, and depurination) [39]; alkylating [40]; oxidative damage [41]; damage caused by low or high LET radiation (single and double strand breaks, SSBs and DSBs, cluster damage); UV-induced damage (photolesions); or damage caused by base analogs and intercalating agents, crosslinking agents, and protein inhibitors, as illustrated in Figure 2 [42,43,44]. Plant ability to respond to DNA damage caused by stress is dependent on various factors, including their anatomy and morphology, behavioral abilities, physiological resilience, phenotypic flexibility, and the effectiveness of DNA repair mechanisms [44,45,46]. Most environmental factors have a complex mode of action and can directly affect the sugar-phosphate backbone of DNA and lead to the formation of SSBs and DSBs or necessitate metabolic activation.

Hydrolytic DNA damage can occur spontaneously or as a result of stress caused by heating, alkylation of bases, or the action of N-glycosylases [44]. Some of these changes lead to the formation of abasic (apurinic or apyrimidinic, AP) sites. In maize root tip cells, spontaneous hydrolytic DNA damage occurs at a frequency of 3.75 × 10^5^ per genome/per cell during the first 20 h of seed imbibition [47]. According to Britt [39], hydrolytic damage is a common event in the plant genome due to constant exposure to oxygen and UV light. Under stress conditions, the incidence of this type of DNA damage considerably increases.

Exposure to high temperatures can result in an accumulation of hydrolytic damage (such as deaminated cytosine and AP sites) and oxidative damage (such as 8-oxoguanine) in the plant genome [48]. However, plants have the ability to effectively cope with a significant amount of DNA damage through mechanisms that include tolerance or an enhanced repair capacity [49].

Oxidative damage in plant cells is one of the best studied phenomena. Sunlight, specifically its most energetic components (low and high LET; UV radiation), along with air pollutants such as ozone, possess enough energy to excite electrons and ionize water molecules. This results in the formation of water radiolysis products (reactive oxygen species, ROS), hydrated electrons (e_aq_^−^), ionized water (H_2_O^+^), hydroperoxyl radicals (HO_2_•), hydroxyl radicals (•OH), hydrogen radicals (H•), reactive nitrogen species (RNS) [49,50], and hydrogen peroxide (H_2_O_2_). These events occur within a very short period of time (~10^−8^ s) [51]. During stress, there is typically an increase in the imbalance between the production and scavenging of ROS [52,53,54]. Recently, two pathways of ROS generation in plant cells have been identified: through plasma membrane-bound nicotinamide adenine dinucleotide phosphate (NADPH) oxidase and peroxidase; or during “electron leak” in chloroplasts, mitochondria, and peroxisomes [55]. Furthermore, different types of ROS have varying levels of reactivity towards DNA. Møller et al. [56] found that among all ROS, only •OH is able to quickly react with DNA molecules. Furthermore, singlet oxygen can also react with DNA, but it primarily does so with guanine. ROS-induced damage to the genome can take many forms, including more than 100 different types of base damage (e.g., single pyrimidine and purine base lesions, inter- and intrastrand crosslinks, purine 5′, 8-cyclonucleosides, DNA–protein adducts, etc.) [57]. Extreme temperatures (frost, cold and heat stress), salinity, drought, water and nutrient imbalance, and high light intensity can trigger oxidative damage in the plant genome [58]. Accumulation of ROS can disrupt the photosynthetic system, carbohydrate metabolism, and some plant cell structures and mechanisms, regardless of whether oxidative damage is caused by endogenous processes or external agents [49,59].

To combat the detrimental effects of oxidative damage, plants have developed a variety of antioxidant regulatory systems, both enzymatic and non-enzymatic [58]. Furthermore, plants possess mechanisms for tolerance, such as ignoring or neglecting induced DNA damage without repair, as discussed by Roldán-Arjona and Ariza [49]. Oxidative stress is often accompanied by nitrosative stress [60]. Activation of RNS, such as NO and nitric dioxide (NO_2_), as well as nonradicals, such as nitrous acid (HNO_2_) and dinitrogen tetroxide (N_2_O_4_), when combined with superoxide radical (O_2_•) can lead to disruptions in lipids, thiols, proteins, and DNA bases [56,61]. These signaling molecules are typically associated with the response to abiotic and biotic stresses, but they also play a crucial role in regulating various processes in plants, such as metabolism, growth and development, solute transport, autophagy, and programmed cell death (PCD) [62]. Despite their importance in plant stress response, their potential role in IR stress has not been fully explored [61]. Primary alkylated DNA lesions are a common occurrence in plant genomes, regardless of their endogenous or exogenous origin. They result from the addition of alkyl groups (such as methyl or ethyl group) to oxygen and nitrogen atoms within DNA bases and phosphodiester bonds [43]. These lesions have genotoxic, cytotoxic, or mutagenic properties because they can obstruct gene transcription and replication process within the plant genome.

Secondary damage to the genome, such as AP sites, DNA strand breaks, and interstrand crosslinks, can also occur [63]. In human genomes, the most common alkylated DNA lesions are N7-methylguanine and N3-methyladenine [43,64]. These types of DNA damage can block replication and transcription, leading to the formation of AP sites [64,65]. AP sites can be produced spontaneously or enzymatically during base excision repair (BER). In Arabidopsis, it has been suggested that different groups of endonucleases are active for the removal of AP sites [65].

The use of alkylation damage in plants is often utilized in breeding and genetic modification programs. There are different types of alkylating agents, which are classified by the number of reactive sites as monofunctional, bifunctional, and polyfunctional [66]. These agents can also be classified by their specific nucleophilic substitution reactions, either as monomolecular nucleophilic substitution (S_N_1) or bimolecular nucleophilic substitution 2 (S_N_2) [67]. S_N_1 agents can affect both nitrogen and oxygen atoms on the bases, while S_N_2 agents mainly affect ring nitrogen atoms (N1, N3 and N7) on the bases. These mutations have been utilized to improve cereal crops and medicinal plants, as well as to breed *Capsicum annuum* in order to obtain economically valuable traits. Examples of S_N_1 alkylating agents include N-methyl-N-nitrosourea (MNU), ethyl methanesulfonate (EMS), and N-methyl-N′-nitro-N-nitrosoguanidine (MNNG), while an example of S_N_2 agents is methyl methanesulfonate (MMS) [68,69].

The main components of solar radiation, namely UV-B and short-wavelength UV-C, induce direct photolesions, such as cyclobutane pyrimidine dimers (CPDs) or pyrimidine (6-4) pyrimidone photoproducts (6-4 PPs), as well as indirect DNA damage through the accumulation of ROS. UV-A radiation primarily causes SSBs, alkali-labile lesions, and oxidative DNA damage due to the low level of absorption by DNA [70].

Exposure to IR, both high and low LET, primarily causes base damage and direct DNA strand breaks, including SSBs and a small number of DSBs (<5%). These breaks occur randomly throughout the plant genome. Depending on the intensity of exposure, AP sites and cluster damage may also occur [71]. It is known that low doses of low LET radiation do not usually cause breaks in the DNA chain. When such breaks do occur, they are more likely to be SSBs than DSBs [72]. SSBs are less severe for the genome as they can be repaired quickly, usually within a minute, as the cell has a copy of intact DNA and can restore its structure during repair. However, if there are more than 100,000 SSBs, cell-cycle arrest may occur. In contrast, DSBs are primarily caused by the breakage of phosphodiester bonds between the sugar residues of two complementary DNA, which occur 10–20 bp apart in both DNA strands [73]. It is known that one to ten DSBs can lead to cell-cycle arrest and ultimately cell death [74]. DNA strand breaks can also be caused by other complex types of abiotic stressors. For instance, heat stress can lead to SSBs and DBSs [48], while cold stress can cause DNA damage, including DSBs in root stem cells [75].

Clustered DNA damage refers to complex DNA injury (including DSBs and non-DSBs) that occurs when there are at least two or more lesions of the DNA helix resulting from single exposure to IR or treatment with radiomimetic agents [76,77]. This type of damage includes harm to DNA bases, SSBs, and AP, as well as modifications to sugar residues [76]. It is believed that cluster damage mainly occurs as an indirect effect of radiation exposure and is highly dependent on the neutralization of free radicals and the structure of chromatin [78,79]. Research on the impacts of IR on plants offers an opportunity to study changes in regions affected by radiation pollution, such as Chernobyl, Fukushima, and the Marshall Islands, as well as in naturally radioactive areas [80,81], to track the adaptation process. Additionally, using radiation in a controlled environment allows radiation mutation breeding to create new plant varieties with desirable traits [82]. In addition to traditional forms of radiation, such as X-rays and gamma rays, there is growing interest in using high-energy particle radiation to enhance the quality of ornamental plants [83], and economically important crops [83,84,85], as well as for algae biofuel production [86].

Plants often need to respond to multiple abiotic factors simultaneously, and the consequences of these impacts can lead to secondary and tertiary stress. The most common secondary stress in plants is oxidative stress (Figure 3), which is caused by an overproduction and accumulation of ROS resulting from an imbalance in internal homeostasis. Heat stress causes a variety of changes in plants, including denaturation of proteins and membrane instability, which in turn cause redox stress and an accumulation of ROS [87]. Exposure to radiation can also contribute to the occurrence of heat stress. Salt stress, caused by high soil salinity, leads to both osmotic and ionic stress. The resulting osmotic stress provokes a decrease in plant water content, which is also known as dehydration stress [88]. Dehydration stress primarily occurs during droughts, but it also occurs as a secondary stress during cold, frost, and heat, as well as due to salt. It also occurs as a tertiary stress following radiation stress [89]. Disturbances in K^+^/Na^+^ homeostasis, a result of ionic stress, can alter plant metabolism, disrupt membrane structure, and change enzyme activity. These changes can lead to an increase in ROS levels, resulting in the development of oxidative stress as a secondary stress [90], as well as nutritional imbalances [88] (Figure 3). Drought and salt stress elicit similar metabolic responses in plants [91].

## 4. Dose–Response Models in Plant Ecotoxicology

Investigating the impacts of chemical agents on plants, known as plant ecotoxicology, can be challenging due to the complexity of interpreting the relationship between the dose of the agent and the observed biological effects. Basic models, such as dose–response models, can aid in understanding this relationship by showing how different doses of a chemical agent can lead to different levels of DNA damage in plants. The impact of chemicals on plants can vary depending on the type of agent, concentration, duration of exposure (acute or chronic)], and the mode of action in which the chemical acts on the plant [92,93]. Furthermore, the choice of plant species, testing method (in vivo or in vitro), and the way the chemical is absorbed and metabolized by the plant cells can also affect the results of ecotoxicological research and the effects of the chemical on plants.

Agents can be classified as genotoxic or non-genotoxic based on how they affect cells [93]. Genotoxic agents cause damage to DNA and proteins, and this damage can be classified as primary or secondary depending on whether or not an inflammatory response is present [94]. Primary genotoxins interact directly with cellular components and DNA and lead to the formation of harmful molecules, such as ROS/RNS. They can damage DNA directly or indirectly through the production of free radicals in the mitochondria and membrane-bound NADPH oxidases [94]. Toxicity tests, both in vivo (using live animals) and in vitro (using cell/tissue cultures of human or animal origin), aim to determine the effective dose of the agent that leads to a carcinogenic effect. Non-genotoxic or epigenetic agents primarily affect cell behavior rather than DNA. These agents include tumor promoters, endocrine modifiers, receptor mediators, immunosuppressants or elicitors, and can cause tissue-specific toxicity and inflammatory responses [95]. Most of the agents have little or no impact on plant health as plants possess mechanisms to prevent the spread of tumor cells [96]. On the other hand, some of these agents can be used to achieve specific effects in plant cells through transformation. A key characteristic of these agents is that they have a threshold dose, below which they do not produce a biological effect [93]. Similarly, plants have a threshold radiation dose of 10 mGy·d^−1^ (417 μGy/h) before adverse effects occur [97]. Genotoxic agents directly damage DNA and chromosomes and have mutagenic effects that do not require metabolic activation. Pro-carcinogenic agents, which do require metabolic activation to cause cancer, also belong to this group. Some inorganic substances, such as metals or metalloids, can also be genotoxic. Their level and distribution in soil and water are crucial for plant growth as some are essential micronutrients [98]. However, excessive amounts could be harmful to both plants and humans. When metabolized by plants, metals/metalloids can enter the food chain and accumulate in the human body, causing harm. High concentrations of these substances in soil also lead to contamination and decreased crop yield [98]. In plant model systems, various agents cause sublethal and lethal effects [92]. These effects often have a specific dose–response relationship, which can be linear or non-linear, with a threshold or non-threshold level. As previously mentioned, non-genotoxic agents typically have a threshold dose, but it has recently been observed that some genotoxic carcinogens also have threshold doses [93].

Haber’s law, one of the earliest models used to describe the relationship between dose and response, states that toxic effects are related to both the concentration and duration of exposure. This model suggests that even small doses of a substance can have an effect, which means that exposure to low levels of a toxic substance over a long period of time can be just as harmful as exposure to a higher level for a shorter period of time. However, in reality, many toxic effects are found to be more influenced by the concentration of the substance rather than the duration of exposure [99]. This means that the amount of a toxic substance in a given environment is more crucial in determining its potential for harm than the length of time when the substance is present. It is important to note that the dose–response relationship is not always linear, and the toxic effects of a substance can also depend on the route of exposure, the organism, and other factors.

Plants, like all biological systems, are complex organisms and their response to external factors depends on various structural and behavioral factors. There are also notable differences in the way plants and animals respond to agents. According to Karban et al. [100], plants tend to have a higher threshold for agents and a lower sensitivity compared to animal models. The relationship between the amount of an agent (a drug or chemical) applied to an organism and the resulting biological response can be represented by a dose–response curve as illustrated in Figure 4. This type of graph illustrates the relationship between the increased dose or the concentration of the agent and the corresponding increase in biological response. There are two main types of dose–response curves, namely graded and quantal. Graded dose–response curves describe the continuous relationship between increased biological response and increased dose or concentration in the single biological unit. These curves are characterized by four parameters as follows: potency (also known as the half-maximal effective concentration or dose, denoted as EC_50_ or ED_50_), which is the dose that produces the maximum effect; slope, which describes how steep the curve is; maximum, which is the highest level of response that can be achieved; and threshold dose, which is the minimum dose required to produce any response [101]. On the other hand, quantal dose–response curves describe the relationship between the proportion of organisms experienced or a not particular effect, known as an “all-or-nothing” phenomenon. Depending on the variation in biological response per unit dose or concentration, different types of dose–response curves can be observed. These can include monotonic curves, where the slope does not change sign; non-monotonic, where the slope changes sign [102]; and curves with no dose–response relationship (WDR), characterized by a zero slope [103]. Examples of different types of curves include linear, non-linear, threshold, sigmoidal, saturation, and U-shaped curves.

In plants, different stressors can produce various dose–response relationships depending on the specific responses and the ability or inability to overcome a particular stressor. Linear response models (Figure 4a,b) with or without threshold dose or concentration (linear no-threshold, LNT; linear threshold, LT) have been used for over 90 years as the standard model for assessing the risk of chemicals, radiation, and environmental agents [107,108]. It should be noted, however, that this model may not always be appropriate for all types of stressors and its limitations should be taken into account. The LNT model, originally developed to explain evolutionary processes [108], has become the main model for assessing the risk of IR effects used by the World Health Organization (WHO) and the Environmental Protection Agency (EPA) as the standard for human health protection. In this model, the biological effect is assumed to be proportional and have a linear or linear-quadratic curve [109]. The linear no-threshold model is based on the principle that radiation is extremely hazardous and there is no safe level of exposure. Even low doses of radiation can result in heritable genetic mutations and tumorigenesis. The target effects of radiation exposure on DNA molecules also follow this pattern [110], which is directly related to the high-dose effects of radiation on human populations. However, this does not necessarily apply to all types of stressors. Furthermore, this model does not always explain the observed short- and long-term effects of low-dose radiation [111], which have been observed in humans during space exploration, after the catastrophic nuclear disasters in Hiroshima and Nagasaki in 1945, and those affected by nuclear disasters at Chernobyl (1986) and Fukushima (2011) [109]. It is worth mentioning that alternative models, such as the hormesis or threshold model, may better explain these effects.

When evaluating the risk for plants, it is important to consider both the potential harm to the plant itself and the potential impact on human health through consumption or other means. For instance, plants grown in areas with high radiation levels or heavy pesticide use may contain dangerous levels of radionuclides or pesticides that could harm humans if consumed [80,112]. Furthermore, doses or concentrations that would affect humans have a greater impact on plants because they have a higher tolerance for radiation and chemical exposure. It should also be noted that various factors, such as the type of agent, the exposure level, the extent of contact with the plant, the accumulation in the plant, the distribution in different plant parts, and the plant ability to neutralize the agent, are crucial in determining the impact. It is also important to consider the context in which the exposure takes place and the specific plant species and variety being studied.

## 5. Abiotic Stress Factors Elicit Off-Target Effects in Plants In Vivo

The wide variety of plant stressors leads to a number of biological effects. Ideally, stressors only affect specific targets, such as the DNA molecule, resulting in targeted effects that can be seen within one or two generations. However, in reality, organism response is more complex and off-target effects, which are not fully understood, are also observed [113]. Despite limited research in this area, radiation exposure often leads to such effects in animal and human model systems, and similar effects are seen in plants due to various environmental stressors. The known off-target effects of radiation exposure, such as signal-mediated effects, stress-induced genomic instability, transgenerational effects, sensitivity to low doses, biphasic response or hormesis, and others, also occur in the plant genome under environmental stress as outlined by the study of Joiner [114]. More research is needed to fully understand the extent and mechanisms of these off-target effects in plants.

## 6. Signal-Mediated Effects in Plants

The signal-mediated effects deviate from the traditional target theory and there is ongoing discussion regarding the dose–response relationship. There are two opposing hypotheses. One states that these effects increase with dose, while the other argues that they do not. Some researchers suggest that the effects reach a saturation point at low doses and no further effects are seen at higher doses [104,115,116]. Others consider that a binary response, with a clear threshold dose below which no effects are observed, is also possible [104]. The shape of the dose–response curve describing these effects is still under investigation and can vary depending on the specific stressor and plant species. Typical dose–response curves describing these effects are shown in Figure 4h. The term “signaling-mediated effects” encompasses two physiological phenomena called bystander and abscopal effects, which refer to the ability of untreated cells to pick up stress signals from treated cells, displaying similar changes [117]. These two effects can be differentiated based on the location of the stress response, either in neighboring cells or in distant organs from the site of stress application (“out-of-field”), respectively [118]. The mechanisms that lead to signaling-mediated effects in plants are still being studied, with some theories suggesting that they result from intracellular gap–junction communication [119] or by extracellular low-molecular-weight factors released in the growth medium, such as ROS, cytokines, calcium ions, and short RNAs [120,121,122,123]. The transmitted signals can be harmful or beneficial to the organism, which are referred to as the “kiss of death” and “kiss of life”, respectively [119]. It is known that even low doses of high and low LET radiation can induce radiation bystander effects. Research on signaling-mediated effects in plants began relatively recently with reports on the occurrence of bystander/abscopal effects in plants [120,121]. In these studies, the researchers showed that post-embryonic developmental defects occur in *Arabidopsis thaliana* after exposure to α-particles and low-energy heavy ions in the shoot apical meristem embryo or intact seeds. Furthermore, some researchers have proposed that the activation of auxin-dependent transcription processes by ROS [124], as well as an increase in homologous recombination events caused by the induction of long-distance DNA damage [125,126], may be responsible for this phenomenon. These findings have been supported by research of *Medicago truncatula* when studying bystander effects [127]. Further examination of the mechanisms underlying bystander effects has revealed changes in DNA methylation patterns and histone modification [128,129], as well as hormone levels [130]. Evidence of chromosomal damage, including rearrangements and sister chromatid exchanges (SCEs), micronuclei, mutations, cell death, altered gene expression [131], and differentiation and changes in microRNA (miRNA) profiles, have also been found [50].

It has been shown that when Arabidopsis plants grown in Petri dishes are irradiated with UV-C and X-ray, communication between plants leads to phenomenon bystander effects [132]. Similar reactions have also been observed in response to various abiotic stress factors, as well as to biotic stressors, such as pathogen infection and pest infestation [133,134]. When one tobacco plant was infected with tobacco mosaic virus, it resulted in an increased frequency of homologous recombination events that spread to uninfected tissue in both a sensitive and resistant cultivar [133]. Other bystander-like effects have been observed in plant–plant interactions when volatile organic compounds emitted by herbivore-damaged cabbage plants trigger the production of defense traits in nearby undamaged plants in field conditions [135].

## 7. Transgenerational Memory Effects in Plants

Plants need to adapt better to constantly changing environments in order to pass on their acquired memory to future generations. This can occur through short-term memory factors, such as stress-induced signaling chemicals, proteins, RNAs, and metabolites in the germline, or through changes in certain epigenetic features, referred to as long-term memory factors [136,137]. Changes in the progeny of plants that occur one or two generations after stress are referred to as intergenerational stress response, while changes that occur in several generation or result in permanent changes, called epimutations, are referred to as transgenerational stress response [137]. The chronic effects of low-dose IR on organisms can result in genetic changes that are inherited by the next generation. Studies on the long-term consequences of nuclear power plant accidents, such as those in Chernobyl and Fukushima, have shown that transgenerational heritable effects of IR on plant germline are particularly evident. These effects include an increase in somatic homologous recombination frequency and activation of the ROS scavenging system in plants [138,139]. There is also an increase in the activity of transposable elements [140]. Factors that contribute to transgenerational memory can include not only different types of IR, but also various biotic and abiotic environmental factors [128,141,142,143,144,145]. Transmission of environmental information through epigenetic marks is crucial for transgenerational memory effects in plants. In recent years, there has been growing interest in studying these phenomena, which can have both negative and positive impacts, such as increased genome instability, higher stress tolerance, and cross-tolerance [146,147].

The epigenetic landscape of a plant that includes DNA methylation [148,149,150,151], post-translational histone modifications [152], and small RNAs [153] can be dynamically altered in response to environmental cues [154]. During cell division, DNA methylation patterns are copied to newly synthesized DNA strands and are heritably transmitted from the parent plants to their offspring. Differences in gene expression or phenotype within and between populations can be attributed to stable “epialleles” where the variation in gene expression is caused by epigenetic differences [151]. A well-studied example of such a phenomenon is “paramutation”, wherein one gene allele can transfer epigenetic information to another allele, resulting in a persistent and inheritable modification in gene expression [148,155]. Epigenetic marks can be removed or maintained during reproduction depending on developmental stage and tissue context of the gametes. For instance, DNA methylation can be erased in male gametes but maintained in female gametes. This differential erasure is called genomic imprinting and can result in distinct epigenetic inheritance patterns across paternal and maternal lineages [156]. Following transmission to the next generation, epigenetic marks are interpreted and translated into changes in gene expression and chromatin structure. One example is DNA methylation, which recruits proteins to modify histones, leading to changes in chromatin and gene expression. These modifications are critical for various developmental processes, stress responses, and adaptation to changing environments. They can be transmitted across generations through mitotic and meiotic divisions, contributing to transgenerational epigenetic inheritance. Histone modifications are now recognized as important players in shaping the phenotype and evolution of plant populations [151,152].

Epigenetic marks have varying capacities for long-term inheritance, with DNA methylation being more stably transmitted than other regulators [156]. While histone modifications are more likely to be reset during meiosis, they can work together with other regulators to ensure transgenerational effects. Transgenerational memory can be induced by coordinated epigenetic regulation driven by various factors, such as histone demethylases, heat-shock transcription factors, and trans-acting siRNA biogenesis, to enhance growth and attenuate plant immunity under increased temperature stress [151].

## 8. Stress-Induced Genomic Instability

Stress-induced instability describes de novo changes at the genomic and chromosomal levels in the progeny of stressed cells, leading to increased carcinogenic susceptibility in these cells. This instability can manifest as an abnormal cell state, such as an accumulation of inherited genomic changes, such as gene mutations, high levels of microsatellite or expanded simple tandem repeat (ESTR) mutation levels, and delayed cell death, or epigenetic modifications, such as changes in DNA methylation, chromatin remodeling, and gene expression [121,157,158]. In addition to genomic instability, stress can also result in chromosomal instability, characterized by high levels of chromosomal rearrangements, formation of micronuclei, increased somatic hyperrecombination, and changes in ploidy levels. Several endpoints have been used to study radiation-induced genome instability in plants, such as analyzing chromosomal aberrations in meiotic pollen mother cells in rice [159], micronuclei formation and cell proliferation in tobacco cells [160], and rearrangements between parental chromosomes in certain root meristem cells in interspecific tobacco hybrids [161]. Significant insights into plant genome instability have been gained through the examination of reconstructed karyotypes of plants produced through IR in breeding practices and analyzed by comet assay [162,163]. It has been proposed that exposure to radiation may cause genome instability, potentially due to an increased accumulation of inherited genomic rearrangements. The presence of specific mutations, such as deletions and translocations, leads to different levels of genomic instability in the irradiated progenitor offspring, which translates into different levels of sensitivity in the lines to IR [162,164]. Initially, a basic understanding of genomic instability in plants came from IR studies; however, further research has revealed that this type of instability can also occur as a result of other methods used in plant breeding, for instance, physical, chemical, or radiation mutagenesis. Recent studies have shown that stress-inducing factors, such as low concentrations of EMS [165], herbicides [166], and salt and heat stress [142], can also lead to genomic instability.

## 9. Bet-Hedging Strategy and Stress-Induced Transgenerational Tolerance of Plants 

Plants have evolved various survival strategies to cope with environmental stress, including bet-hedging and transgenerational stress tolerance. These strategies increase phenotypic diversity and endurance, allowing plants to adapt and survive in changing environments [147]. Bet-hedging can occur through an intergenerational nonheritable effect, where the phenotype of the progeny is influenced by the environmental conditions experienced by the parent [137]. Otherwise, bet-hedging can occur through a transgenerational heritable effect, where the phenotype of the offspring is influenced by the natural surroundings experienced by previous generations [167]. For instance, the Cape Verde islands accession (Cvi-0) of Arabidopsis shows increased phenotypic diversity and endurance in response to high temperatures across two successive generations. The resulting offspring are better adapted to survive in harsh environments. S1 and S2 generations exhibit desirable traits, such as increased plant height, length, and more rosette leaves [168], but these characteristics may diminish in subsequent generations [137]. Similarly, transgenerational stress tolerance in plants refers to their ability to endure various degrees of environmental stress [169], primarily determined by the parental exposure to stress, known as priming stress [136]. Different environmental stressors can induce a phenomenon called primed state and tolerance [169,170,171,172]. These stressors can be applied during various stages of plant development, from seed germination to maturity. Several plant systems have been identified as causes for transgenerational plant tolerance, including hormonal signaling pathways, antioxidant defense systems, and epigenetic modifications [173]. The key aspect of this tolerance is an enhancement in plant growth, yield, performance, and defense [169], which is epigenetically controlled and can be sustained over multiple generations free of stress [173]. This enables plants to not only endure challenging environments but also to maintain an advantage over other individuals, while also increasing their overall fitness and chances of long-term survival [174].

## 10. Cross-Tolerance

Many studies focus on explaining the effects on plants after being exposed to a single stressor under controlled environmental conditions. However, these effects in natural and field environments are a result of a combination of various biotic and abiotic stressors [175]. Cross-tolerance (cross-resistance or cross-protection) is a phenomenon in which preliminary plant exposure to a mild primary stressor induces tolerance to subsequent exposure to another, more severe stressor. There are three types of cross-tolerances in plants as follows: 1. Transcriptional overlap between different stress responses [2,175]; 2. Induced cross-tolerance, in which the application of priming stress or suppression stress activates plant stress memory and makes it tolerant to subsequent exposure to a different stress [175,176]; and 3. Inherent cross-tolerance, which is characterized by genetic overlap between different stressors. The combined effect of two or more stressors can have a positive or negative impact on plant development and metabolism [177]. Some stress factors may have similar modes of action, using the same signaling molecules and mechanisms [178].

Due to biological plasticity, plants have the ability to readjust and enhance their metabolism and quality characteristics. In this process, the disruption homeostasis of ROS, RNS, and RCS (reactive carbonyl species) in plants is crucial. Severe stress can cause an overproduction of ROS (such as H_2_O_2_), leading to oxidative stress. In mild stress or in severe but short-term stress, ROS function as signaling molecules and trigger tolerance mechanisms in plants [2,175,176,178]. Other signaling molecules from the RNS family (e.g., NO) and RCS (e.g., methylglyoxal) play similar roles in building plant tolerance [176,178]. They lead to synergistic co-activation of glyoxalase and antioxidant systems in plants. Tolerance mechanisms also include the accumulation of plant hormones, osmolytes, and heat-shock proteins (HSPs), which contribute to the protection of cellular components, membranes, and proteins by restoring osmotic balance in cells or serving as chemical messengers in signal transduction pathways [7,179]. The cellular signal transduction network involves important players, such as HSPs, heat-shock transcription factors (HSFs), and mitogen-activated protein kinases (MAPKs), which yield specific signaling and connections between different types of stressors [176,178]. To help plants tolerate stress, the expression of stress-responsive genes is controlled by adjusting the levels of specific miRNAs, a process known as post-transcriptional gene regulation [178].

Phytohormones, including salicylic acid (SA), ethylene (ET), jasmonate (JA), abscisic acid (ABA), auxin (AUX), brassinosteroid (BR), gibberellic acid (GA), cytokinin (CK), and strigolactones (SLs), work together to activate defense gene expression and orchestrate effective plant defense responses against abiotic and biotic stress [180,181]. Abscisic acid (ABA) is considered the master hormonal switch that determines whether to prioritize abiotic or biotic stress responses based on the specific nature of the stressors [24]. ABA primarily regulates plant responses to drought, low temperature, and salinity, but it also mediates defense against pathogens [179,182,183]. Jasmonic acid (JA) enhances resistance against hemibiotrophic pathogens, improves tolerance to abiotic stress, and plays a crucial role in plant response to a range of stressors, including drought stress, ozone stress, UV stress, salinity stress, and cold and temperature stress [184]. JA interacts with other hormone signaling pathways, such as auxin, ethylene (ET), ABA, salicylic acid (SA), brassinosteroids (BRs), and gibberellin (GA), suggesting that JA may also function as a central signal in the network of phytohormones. Salicylic acid is usually involved in the regulation of pathogen-associated protein expression, but it also plays an important role in the response to abiotic stresses, including drought, low temperature, and salinity stress [185,186]. When plants are exposed to multiple stressors, such as drought and heat, they often exhibit increased levels of ethylene, which promotes the expression of stress-responsive genes and enhances plant ability to withstand multiple stressors [187]. Thus, cross-talk between phytohormones is a vital component of plant defense responses under abiotic and biotic stress.

## 11. Low-Dose Hyper-Radiosensitivity and Radioresistance

The concept of low-dose hyper-radiosensitivity is well-established in mammalian systems, where cells display resistance to high single doses of radiation but show sensitivity to small single doses (Figure 4g). This is typified by a limited number of exposures, such as high and low LET IR and chemotherapy drugs [114]. However, experiments on low-dose hyper-radiosensitivity in plant models have not been adequately described [114]. Eriksson [188] is one of the few researchers to do so, reporting this phenomenon in irradiated maize plants after exposure to a dose of 50 cGy where the frequency of mutation induction and lethality in pollen grains was higher than the spontaneous mutation rate. The differences in radiation response at low and high doses of radiation are thought to be due to the different sensitivity of the cell-cycle phases [189]. Further research is needed to fully understand the mechanisms underlying low-dose hyper-radiosensitivity in plant models and to examine the implications of this phenomenon for plant breeding and crop production.

## 12. Biphasic Dose–Response Effects in Plants: Hormesis, Stress-Induced Priming, and Adaptive Response

Although the existence of biphasic dose–response effects in plants has been known since Darwin’s time, it has long been incorrectly associated with homeopathy, causing a stagnation in its study. Another reason for the lack of sufficient knowledge is that this phenomenon has been studied by different disciplines. For many years, it was rejected by governmental regulatory agencies because it contradicts the established dose–response approach to risk assessment [190]. Recently, Calabrese and Agathokleous [190] reported that there are over 30 different terms used to describe the biphasic dose–response model, including U-shaped, adaptive responses, hormesis, priming, preconditioning, and others, which all describe different aspects of the same phenomenon [191].

In plant studies, various terms have been employed based on the type of stressors. For instance, Ancel and Lallemand [192] used the term “preconditioning” to refer to this phenomenon in plants following X-ray irradiation. As noted in the review by Calabrese and Baldwin [193], the concept of “chemical hormesis” can be traced back to the studies from the late 19th century, which demonstrate the stimulatory effects of sodium hypochlorite on seed germination and the influence of different metals on root growth. In the mid-1970s, the term “adaptive response” was utilized to describe the same phenomenon that occurs after chemical mutagens, and later in the 1980s, it was also used for IR [194]. When defense mechanisms were activated as a result of pathogens, arthropod attacks, or adverse environmental conditions, the term “defense priming” was introduced [195,196]. In recent years, this phenomenon has gained increased attention, leading to its deeper understanding.

Hormesis is considered a quantitative estimate of biological plasticity [197]. The basis of hormetic response is prior exposure to a low dose or concentration of a stress trigger (“priming” stress), which can reduce the toxic effects of subsequent exposure of a higher dose or concentration (“challenge” stress) of the same or a different stress trigger [194,198,199]. The hormetic dose–response curve is often depicted as a U-shaped or J-shaped curve, with the main features being the hormetic stimulatory zone (HSZ) with subinhibitory doses; the maximal stimulatory dose (MSD), which is the percentage change from the control dose (usually <200% of control response) [200]; and the selection of the no observable adverse effects level (NOAEL), or the zero equivalent point (ZEP) or thresholds, followed by inhibitory doses where adverse effects are observed [201,202].

The effects of hormesis on organisms can be either harmful, known as distress, or beneficial, known as eustress [202,203]. In plants, there are two main types of hormetic models, namely inverted U-shaped and U-shaped (Figure 4d,e). The inverted U-shaped curve describes a response in which low dose increases and high dose decreases plant growth and photosynthesis parameters, genotoxicity, and mutagenesis [204]. The U-shaped curve, on the other hand, shows a reduction in adverse effects at low doses and an enhancement of adverse effects at high doses, as can be observed in defense mechanisms such as activities of the major scavenging enzymes, such as ascorbate peroxidase (APX), guaiacol peroxidase (GPX), superoxide radicals, endo-proteinase isoenzymes, carbonyl and malondialdehyde groups, etc. [204]. A third type of hormesis dose–response curve with two dents has also been proposed, but it is specific to plants and there is limited evidence of its existence. This model has been observed in different plants under heavy metal stress [205,206]. The scientific literature is abundant with research on the hormesis behavior of plants, which is triggered by various stress factors. These effects are observed at different levels of biological organization, including cells, organs, organisms, and communities [107]. Hormetic responses have been observed in plants following exposure to a wide range of agents that affect plant growth and development, such as macro- and micronutrients [207,208], biostimulants [209], herbicides and fungicides [210,211,212], heavy metals and metal ions, nanoparticles [204,213,214,215,216], temperature [200], phytohormones [217], heat stress [218], light [219], and pathogens [195].

A growing body of research has shown that both IR and non-IR exposure can have hormetic effects on plants. In general, hormesis is associated with pretreatment of plants with relatively weak exposure (called conditioning clastogenic dose), which increases their resistance to radiation, followed by exposure to higher doses (or challenge dose) of the agents some hours later [220]. Hormesis can be induced by both low and high doses of radiation [221,222,223]. Typically, radiation hormesis in plants has a positive effect, resulting in increased germination, growth rate, height, weight, pigment content, flowering, fertility, accelerated development, and increased radiation resistance. The degree of hormesis depends on the genetic characteristics of the seeds or plant, moisture of the seeds, type of low-dose radiation, and duration of irradiation [224,225]. The adaptive response triggered by hormesis includes both short-term mechanisms, such as the use of existing proteins, and long-term mechanisms, such as the expression of genes encoding specific enzyme systems. Activation of HSPs, proteasomes, and kinase cascades can also occur [225,226]. During hormesis, several mechanisms are activated in plants, including the detoxification of ROS through increased levels of ABA, followed by increased levels of H_2_O_2_, activation of DNA repair mechanisms, removal of damaged cells through apoptosis, alteration of nitrogen metabolism, and stimulation of immune response [208,222,226,227,228]. The hormetic part of the adaptive response is associated with permanent genetic or epigenetic changes [229]. Recent studies suggest that epigenetic mechanisms play a role in plant adaptation and generation of transgenerational memories to stress [230]. It is worth mentioning that the complexity of hormesis requires additional research to fully comprehend its underlying mechanisms.

## 13. Conclusions and Perspectives

In summary, research into the responses of plants to stress is rapidly growing and many questions still need to be explored. The study of how plants handle stressors is crucial, not only for agriculture and the environment, but also for the survival of all living organisms. We summarized current knowledge on the direct effects of high doses or concentrations of stressors on plants (target effects), as well as the effects seen at low doses or concentrations (non-target effects), with a specific focus on the low-level effects that deviate from traditional linear models and do not have a clear threshold in the dose–response relationship. Prolonged exposure to stressors enables plants to adapt and become more resilient, resulting in increased resistance. The response of plants to stress factors is influenced by lifestyle and, in general, plants are more tolerant and able to withstand a wider range of stressors than animals or humans. This is due to the various developmental, physiological, biochemical, genetic, and epigenetic strategies they employ to overcome stress conditions.

The information provided will assist plant scientists in identifying and investigating these phenomena to gain a deeper understanding of the mechanisms behind plant stress response. This response is a fundamental aspect of the natural world and understanding it can yield predictions about the effects of excessive chemical use in agriculture and the potential impacts of untested substances. By considering these factors, scientists and policymakers can develop effective strategies for mitigating the effects of climate change on plants and preserving their habitats and ecosystems.

## Figures and Tables

**Figure 1 ijms-24-05105-f001:**
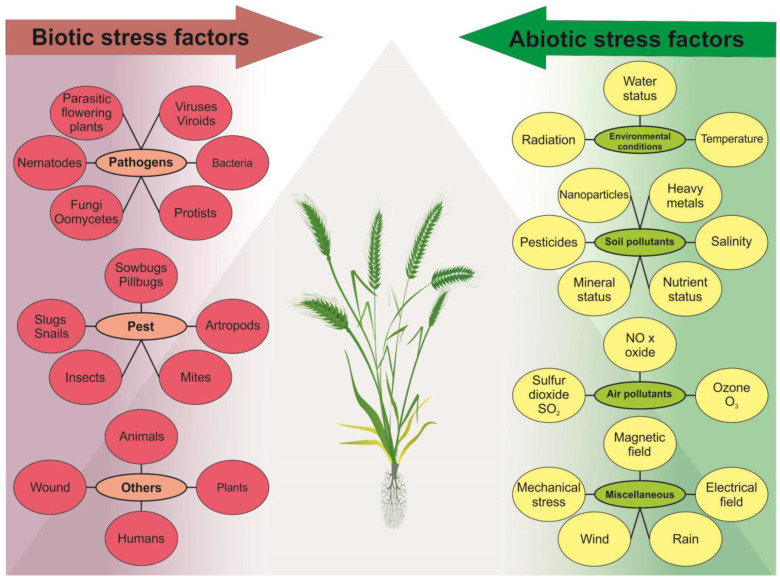
Plants can be subjected to stress due to both biotic factors, such as pathogens, pests, and direct animal and human influences, and abiotic factors, such as diverse climatic changes, soil and air pollution, and magnetic fields. These stressors can act individually, simultaneously, or in various combinations, and disrupt plant homeostasis, impede growth, and impact development in wild populations and cultivated field crops.

**Figure 2 ijms-24-05105-f002:**
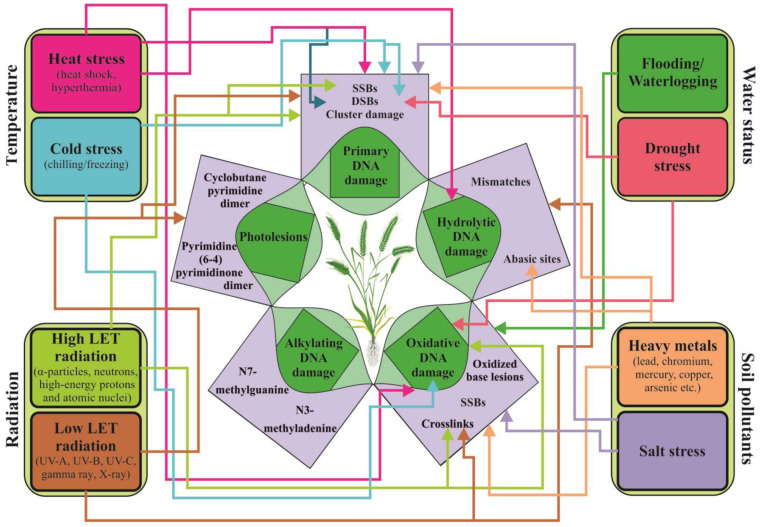
Overview of the main types of DNA lesions observed in plant genomes in response to abiotic stress. Radiation as a stressor can lead to the accumulation of various types of DNA damage in plant cells, including direct DNA damage (SSBs, DSBs, and cluster damage), photolesions (6-4 photoproducts and cyclobutane-pyrimidine dimers), base damage, and interstrand crosslinks, depending on the radiation properties (high or low LET). Temperature, water, and salt stress can cause direct DNA lesions and oxidative damage due to an overproduction of ROS. Heat stress can also lead to hydrolytic lesions, similar to those caused by heavy metal stress. Primary alkylated damage is not connected to the stress factors shown in the figure as it results from the application of alkylating agents, which are not typical stress factors for plants. The relationships between specific DNA lesions and stress factors are indicated by arrows.

**Figure 3 ijms-24-05105-f003:**
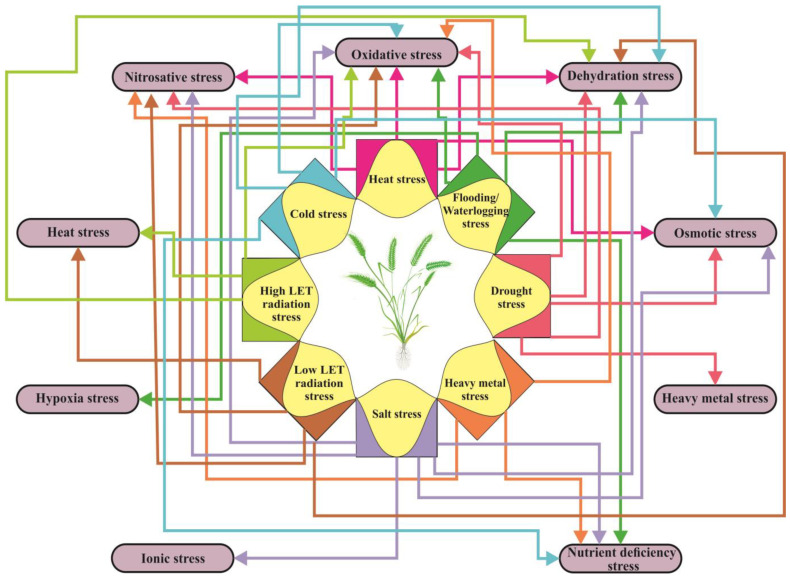
A simplified illustration of the interaction between plant responses induced by primary abiotic stress factors and those caused by secondary and tertiary stressors. The primary stress factors are indicated in the inner section of the figure, while the secondary and tertiary stressors are shown in the outer section. The arrows depict the relationship between the various stressors.

**Figure 4 ijms-24-05105-f004:**
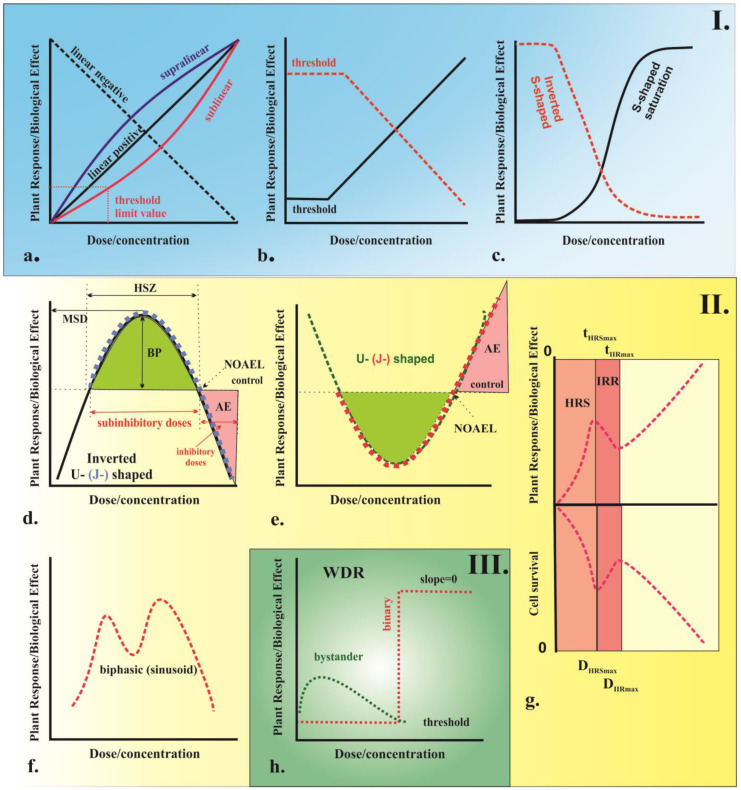
Hypothetical curves of the dose–response relationship. I. Monotonic type (blue background). (**a**) Linear, sublinear and supralinear curves with no clear threshold. The lowest dose at which an effect can be observed is named the threshold limit value (TLV); (**b**) threshold curves; (**c**) S-shaped curves (saturation curve). The S-shaped curves are characterized by an increasing trend, reflecting the beneficial effects on the plant organism as the dose/concentration gradually increases. II. Non-monotonic (yellow background). (**d**) U-shaped or J-shaped curves; (**e**). inverted U- or β-shaped curves. This type of curve reflects the hormetic response at low doses; (**f**) biphasic curve; (**g**) N-shaped dose–response curves of HRS/IRR response in terms of DSBs and cell survival (adapted from Thomas et al. [104] and Devic et al. [105]). III. Curves without dose–response relationship (WDR) (green background). (**h**) Binary curve and bystander response (adapted from Prise and O’Sullivan [106]).

## Data Availability

Not applicable.

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
