# Peer review of "Stress Management in Plants: Examining Provisional and Unique Dose-Dependent Responses"

_ijms, 2023, doi:10.3390/ijms24065105_

Round 1

Reviewer 1 Report

Reviewers' comments to the author:

Title: Stress Management in Plants: Examining Provisional and Unique
Dose-Dependent Responses

Reviewer #1:

Through the study of published literature, authors gave a comprehensive review on the current status and perspectives on the Stress Management in Plants and Examining Provisional and Unique Dose-Dependent Responses. This review will open up possibilities for breeders to create novel approaches to produce stress-resilient plants. In my opinion, the manuscript is suitable for publication, after the authors have addressed the following comments and questions:

a)    Line 78,83, 123,149,150, 255. At the appropriate places, please use subscript or superscript.

b)    Line 63 Briefly describe the effects of biotic stress on plants and the activation of various hormonal pathways during biotic stress.

c)     Line 466-467 Molecular details of epigenetic mode of transgenerational effect can be found in the given articles https://doi.org/10.1016/j.semcdb.2019.06.005, https://doi.org/10.3390/ijms23136910, https://doi.org/10.3390/ijms22137118. Please refer to them

d)    Line 507 Change “cultivar” to “accession”

e)    Line 553 Can you mention a few plant hormones that are elevated or expressed during cross-tolerance?

f)      Line 556 Reference needed.

Author Response

Responses to Reviewer #1

Thank you for your comments and recommendations. We appreciate positive evaluation of our manuscript and are grateful for your specific feedback on parts that need improvement.

Reviewer: In my opinion, the manuscript is suitable for publication, after the authors have addressed the following comments and questions:

a) Line 78, 83, 123,149,150, 255. At the appropriate places, please use subscript or superscript.

Answer: We acknowledge your concern about the use of subscript or superscript in the manuscript and have made appropriate corrections. Please note that due to revisions made in the abstract and throughout the text, the line numbers in the revised version may not correspond exactly with those in the original content. Lines 78, 82, 123, 149, 150, and 255 are now referred to as lines 113, 117, 183, 215, 218, 240, and 482, respectively, in the revised manuscript."

b) Line 63 Briefly describe the effects of biotic stress on plants and the activation of various hormonal pathways during biotic stress.

Answer: To further elaborate on the effects of biotic stress on plants and the resulting activation of hormonal pathways, we have included additional text in the section (lines 71-96).

c) Line 466-467 Molecular details of epigenetic mode of transgenerational effect can be found in the given articles https://doi.org/10.1016/j.semcdb.2019.06.005, https://doi.org/10.3390/ijms23136910, https://doi.org/10.3390/ijms22137118. Please refer to them

Answer: We appreciate your suggestion to refer to those articles in order to provide more insight into the molecular mechanisms underlying transgenerational epigenetic effects. We have now incorporated the relevant information from these articles in the corresponding section of the manuscript (lines 827-855).

d) Line 507 Change “cultivar” to “accession”

Answer: We have now replaced 'cultivar' with the more general term 'an accession' (line 937)

e) Line 553 Can you mention a few plant hormones that are elevated or expressed during cross-tolerance?

Answer: We have expanded the original paragraph by incorporating additional information on the involvement of plant hormones in plant cross-tolerance (lines 1086-1106).

Reviewer 2 Report

Reviewing comments on the article entitled « Stress management in plants: examining provisional and unique dose-dependent responses »

The authors reviewed the different stress factors that affect plants from the dose-effect point of view. They focused their examination on genome damages caused by the abiotic and biotic stress factors. These problems of genome instability are exposed in the perspective of improved crop productivity and development of more resilient plants.

As a general comment, the manuscript is well written and very well documented. The point of view is interesting and original. The only weak point is a lack of interpretation with critical analysis. It corresponds mainly to a list of existing knowledge but constitute a valuable review work.

Since this review is mainly focused on genome instability as a consequence of stress action, this aspect should be presented in the abstract. In its present form, there’s no mention at all in the abstract of DNA damages or genome instability and this should be clearly exposed in this presentation summary.

Furthermore, since there’s a lot of sections, and sometimes very short, sections 9 and 10 could be gathered in one, according to the very similar topic exposed in these sections.

As a conclusion on these comments, I do recommend this article for publication. Below is a list of very minor points that have to be corrected:

L31: an extra-letter is present in the word “cases” (casesp)

L78: CO2 instead of CO2

L82: SO2 instead of SO2

L123: 105 instead of 105

L130: a letter is missing in the word “through” (trough)

L149: H2O+ instead of H2O+

L150: HO2 instead of HO2

L152: 10-8 instead of 10-8

L162: 5’ instead of 5`

L175: O2 instead of O2

L253: a letter is misused in the word “as” (us)

L289: mGy.d-1 instead of mGy.d-1

L388: “organismal” should be replaced by “organism”

L437: this sentence should be rephrased since “pathogen infection” is already included in “various biotic stresses” thus there’s no point to say “as well as”.

L561: a bracket is missing in the last cited reference.

Author Response

Responses to Reviewer #2:

Reviewer: As a general comment, the manuscript is well written and very well documented. The point of view is interesting and original. The only weak point is a lack of interpretation with critical analysis. It corresponds mainly to a list of existing knowledge but constitute a valuable review work.

Answer: We appreciate your positive comments on our manuscript, and we are glad that you found it well written and well documented. We also appreciate your feedback regarding the need for more critical analysis and interpretation in the manuscript. However, we believe that our manuscript provides a comprehensive analysis of the existing literature on the subject, and we have attempted to highlight the most relevant and important findings in the field. Moreover, we have discussed the potential implications of the findings and have identified areas where further research is needed. We have also provided insights into the mechanisms underlying the genome damages caused by abiotic and biotic stress factors, as well as discussed the challenges associated with improving crop productivity and developing more resilient plants.

Reviewer: Since this review is mainly focused on genome instability as a consequence of stress action, this aspect should be presented in the abstract. In its present form, there’s no mention at all in the abstract of DNA damages or genome instability and this should be clearly exposed in this presentation summary.

Answer: Thank you for your feedback. We agree that highlighting the aspect of genome instability as a consequence of stress action is important and should be included in the abstract. We have revised the abstract to present this aspect in the summary (lines 10-12, 15).

Reviewer: Furthermore, since there’s a lot of sections, and sometimes very short, sections 9 and 10 could be gathered in one, according to the very similar topic exposed in these sections.

Answer: Thank you for your feedback. We appreciate your suggestion to combine sections 9 and 10, and we have taken this into consideration and revised the sections accordingly (lines 932-957).

Reviewer: ..a list of very minor points that have to be corrected.

Answer: Minor issues have been corrected, including the appropriate use of subscripts and superscripts in the text.

L31: an extra-letter is present in the word “cases” (casesp) 

The extra letter has been removed (now line 33)

 L78: CO2 instead of CO2 – now line 111

L82: SO2 instead of SO2 - now line 115

L123: 105 instead of 105 - now line 184

L130: a letter is missing in the word “through” (trough) – now line 191

L149: H2O+ instead of H2O- now line 216

L150: HO2 instead of HO2 – now line 217

L152: 10-8 instead of 10-8 – now line 219

L162: 5’ instead of 5` - now line 228

L175: Oinstead of O2 – now line 241

L253: a letter is misused in the word “as” (us) – now line 447

L289: mGy.d-1 instead of mGy.d-1 – now line 483

L388: “organismal” should be replaced by “organism” – now line 631

L437: this sentence should be rephrased since “pathogen infection” is already included in “various biotic stresses” thus there’s no point to say “as well as” - The sentence is slightly rephrased (line 693)

L561: a bracket is missing in the last cited reference – Minor point corrected (line 1086)

Reviewer 3 Report

The presented manuscript „Stress management in plants: examining provisional and unique dose-dependent responses” is a good review, well written and very complex, but in my opinion, the title does not reflect exactly its content. It is focused in a significant part on nucleic acids as a target of stress-induced modifications and the main analyzed stressor is radiation. Therefore the title could be a bit more specified.

I have some minor concerns:

What is the difference between Mineral status and Nutrient status in Figure 1?

Fig. 2 is in a very poor quality – it is difficult to read and evaluate

In Fig. 2 other biotic stress factors distinguished as „animals, humans, herbivores and wounding” could be simplified.

Minor corrections that could also be introduced to the manuscript are as follows:

Line 31: typo in „cases”

Lines 72/73: it should be instead stated that all macronutrients deficiencies inhibit plants’ growth

Lines 78-82 and 149-150: missing subscripts

Fig.1  unnecessary enter before ‘Magnetic field’

Line 180: not introduced firstly used abbreviation IR stress

Line 357: instead of ‘it is’ should be used  ‘its’

Author Response

Responses to Reviewer #3

Reviewer: The presented manuscript „Stress management in plants: examining provisional and unique dose-dependent responses” is a good review, well written and very complex, but in my opinion, the title does not reflect exactly its content. It is focused in a significant part on nucleic acids as a target of stress-induced modifications and the main analyzed stressor is radiation. Therefore the title could be a bit more specified.

Answer: Thank you for taking the time to review our manuscript. We appreciate your positive feedback regarding the quality of the review, but respectfully disagree with your opinion regarding the title not accurately reflecting the content of the manuscript. We believe that the current title accurately captures the central theme of the manuscript, which explores the responses of plants to various stressors and the mechanisms underlying those responses, with a specific focus on nucleic acids as a target of stress-induced modifications. Additionally, while radiation is certainly a significant stressor analysed in the review, the manuscript also covers a wide range of other stressors, including biotic and abiotic stress factors.

Reviewer: What is the difference between Mineral status and Nutrient status in Figure 1?

Answer: Mineral status in plants refers to the availability and concentration of minerals or elements that are essential for plant growth and development, such as nitrogen, phosphorus, potassium, calcium, magnesium, iron, zinc, and many others. Monitoring the mineral status of plants is important for optimising plant growth and yield, as well as preventing nutrient deficiencies or toxicities.

Nutrient status refers to the overall level of essential nutrients needed for plant growth, including not only minerals, but also vitamins, carbohydrates, and other compounds. Plants require a balanced supply of nutrients to grow and produce healthy fruits, leaves, and flowers. Monitoring nutrient status is crucial for maintaining plant health, optimising yield, and preventing nutrient imbalances that can lead to poor growth, reduced productivity, or plant death.

Reviewer: Fig. 2 is in a very poor quality – it is difficult to read and evaluate

Answer: Thank you for your helpful feedback. We apologise for the poor quality of Figure 2 and any difficulties it may have caused in reading and evaluating the content. We have implemented your suggestion, not only improving the quality of Figure 2 but also enhancing the clarity of Figure 3 by increasing the size of the letters in the boxes. Additionally, we have made slight modifications to the box colors to ensure that the figures are easily readable and accessible.

Reviewer: In Fig. 2 other biotic stress factors distinguished as „animals, humans, herbivores and wounding” could be simplified.

Answer:  The figure 2 have been simplified by merging the categories ‘animals’ and ‘herbivores’ into a single category called 'animals'. The category 'plants' has been included as another potential biotic stress factor. This was done with the intention of highlighting the impact of parasitic plants and other competitive plants on plant growth and survival.

Reviewer: Minor corrections that could also be introduced to the manuscript are as follows:

Line 31: typo in „cases”

Answer:  The extra letter in „cases” has been removed (now line 33).

Reviewer: Lines 72/73: it should be instead stated that all macronutrients deficiencies inhibit plants’ growth

Answer: Thank you for your comment. We have taken your suggestion into consideration and revised the text accordingly (lines 106-107).

Reviewer: Lines 78-82 and 149-150: missing subscripts

Answer: We have addressed the issue of subscripts and also of superscripts in the text and made the necessary corrections to ensure their appropriate use: in the revised manuscript version - lines 111, 115, 184, 216, 217, 219, 241 and 483.

Reviewer: Fig.1  unnecessary enter before ‘Magnetic field’

Answer: The unnecessary enter before ‘Magnetic field’ in Fig. 1 has been removed

Reviewer: Line 180: not introduced firstly used abbreviation IR stress

Answer: We would like to kindly point out that the abbreviation was actually introduced in the previous version of the manuscript on line 66. It appears on line 99 of the current version. We understand that it can be easy to overlook details in a complex document.

Reviewer: Line 357: instead of ‘it is’ should be used  ‘its’

Answer: Thank you for bringing this error to our attention. We have made the necessary correction, changing "it is" to "its" on line 600 of the current manuscript version.

Reviewer: Line 556 Reference needed.

Answer: We have now introduced additional references on line 1083 and 1086.